# Effects of injectable contraception with depot medroxyprogesterone acetate or norethisterone enanthate on estradiol levels and menstrual, psychological and behavioral measures relevant to HIV risk: The WHICH randomized trial

Mandisa Singata-Madliki[1☯], Jenni Smit[2], Mags Beksinska[2], Yusentha Balakrishna[3], Chanel Avenant[4], Ivana Beesham[2]*, Ishen Seocharan[3], Joanne Batting[1], Janet P. Hapgood[4,5☯], G. Justus Hofmeyr[1,6,7☯]

1 Effective Care Research Unit, Eastern Cape Department of Health/Universities of the Witwatersrand and Fort Hare, East London, South Africa, 2 Wits MRU (MatCH Research Unit), Department of Obstetrics and Gynecology, Faculty of Health Sciences, University of the Witwatersrand, Durban, South Africa, 3 Biostatistics Research Unit, South African Medical Research Council, Durban, South Africa, 4 Department of Molecular and Cell Biology, University of Cape Town, Cape Town, South Africa, 5 Institute of Infectious Disease and Molecular Medicine, University of Cape Town, Cape Town, South Africa, 6 Walter Sisulu University, East London, South Africa, 7 Department of Obstetrics and Gynecology, University of Botswana, Gabarone, Botswana

☯ These authors contributed equally to this work.
* i_beesham@yahoo.com

# Abstract

## Background

Observational data suggest lower HIV risk with norethisterone enanthate (NET-EN) than with depo-medroxyprogesterone acetate intramuscular (DMPA-IM) injectable contraceptives. If confirmed, a switch between these similar injectable methods would be programmatically feasible and could impact the trajectory of the HIV epidemic. We aimed in this paper to investigate the effects of DMPA-IM and NET-EN on estradiol levels, measures of depression and sexual activity and menstrual effects, relevant to HIV risk; and to ascertain whether these measures are associated with estradiol levels.

## Methods

This open-label trial conducted at two sites in South Africa from 5 November 2018 to 30 November 2019, randomized HIV-negative women aged 18–40 to DMPA-IM 150 mg intramuscular 12-weekly (n = 262) or NET-EN 200 mg intramuscular 8-weekly (n = 259). Data were collected on hormonal, behavioral and menstrual effects at baseline and at 25 weeks (25W).

**Data Availability Statement:** The de-identified dataset is available on the South African Medical Research Council portal via the following link: https://medat.samrc.ac.za/index.php/catalog/51.

**Funding:** This is an investigator-initiated, academic, non-commercial study funded by the South African Medical Research Council Grants Innovation and Product Development (SAMRC N/A grant number) to MS, including hosting the randomization program and providing data management and statistical support. The SAMRC statistician (YB) and data manager (IS) were members of the study team. Other than the trial statistician and data manager, the funder had no other role in the study.

**Competing interests:** The authors have no competing interests to declare that are relevant to the content of this article.

## Results

At 25W, median 17β estradiol levels were substantially lower than at baseline (p<0.001) for both methods: 76.5 pmol/L (interquartile range (IQR) 54.1 to 104.2) in the DMPA-IM group (n = 222), and 69.8 pmol/L (IQR: 55.1 to 89.3) in the NET-EN group (n = 225), with no statistical difference between the two methods (p = 0.450). Compared with DMPA-IM, NET-EN users reported significantly less amenorrhoea, fewer sexual acts, fewer users reporting at least one act of unprotected sex, more condom use with steady partner, more days with urge for sexual intercourse, more days feeling partner does not love her, and more days feeling sad for no reason. We did not find a clear association between estradiol levels and sexual behavior, depression and menstrual effects. Behavioral outcomes suggest less sexual exposure with NET-EN than DMPA-IM. The strength of this evidence is high due to the randomized study design and the consistency of results across the outcomes measured.

## Conclusions

Estradiol levels were reduced to postmenopausal levels by both methods. Secondary outcomes suggesting less sexual exposure with NET-EN are consistent with reported observational evidence of less HIV risk with NET-EN. A randomized trial powered for HIV acquisition is feasible and needed to answer this important question.

## Trial registration

PACTR 202009758229976.

## Introduction

Access to effective and safe contraception is critical to empowerment and well-being of individuals and to prevent the burden of unintended pregnancies. Non-barrier contraception might increase HIV risk by reducing motivation for condom use, but also avoids the likely increased risk of HIV acquisition during pregnancy [1, 2]. For individuals who require effective contraception, it is the relative HIV risk associated with available methods that is of importance.

About 38% (16.5 million) of modern contraceptive users in sub-Saharan Africa use progestin-only injectables [3], predominantly the three-monthly, intramuscular injection of 150mg depot medroxyprogesterone acetate (DMPA-IM) [4]. Norethisterone enanthate (NET-EN), a two-monthly, intramuscular injection of 200mg NET-EN, is also widely used in South Africa [5]. Two non-randomized head-to-head comparisons of HIV risk among participants using DMPA-IM versus NET-EN indicated a potential 32–40% greater risk of HIV acquisition for DMPA-IM users versus NET-EN users [6–9], while one found no difference [10]. More recently, HIV acquisition among vaginal lactobacillus-dominant participants using DMPA-IM was reported to be 3-fold that of those using NET-EN [11]. Given the potential for confounding factors in these observational studies, a definitive answer on the relative HIV risks of DMPA-IM versus NET-EN remains elusive. The issue is particularly important because if a lower risk with NET-EN than DMPA-IM were to be confirmed, a switch between these popular intramuscular methods would be programmatically feasible.

Another approach to gaining insights into the relative risks and benefits of DMPA-IM versus NET-EN is to compare biological and behavioral data relevant to HIV acquisition or other

side-effects, within the context of a randomized trial. The effects of hormonal contraception on HIV acquisition may include effects both reducing risk (e.g. reduced coital activity due to reduced libido/sense of wellbeing and reduced coitus during menstruation associated with oligo-amenorrhoea) and those increasing risk (e.g. immunological effects, effects on barrier function and microbiome in the female genital tract (FGT), and hypoestrogenism) [9, 12–16]. In a previous randomized trial, we found reduced coital activity among participants randomized to injectable progestogens versus the copper intrauterine device (IUD) [17, 18]. The Evidence for Contraceptive Options and HIV Outcomes (ECHO) trial also found reduced condomless coital activity and coitus during menstruation among participants allocated to hormonal contraception than to the copper (Cu) IUD [19]. DMPA-IM was associated with significantly less coital exposure than the levonorgestrel (LNG) implant with respect to: multiple sex partners, new sex partner in the last three months, any unprotected sex, and no condom used for last sex act. These self-reported outcomes were supported by objective data from an ancillary study at three of the ECHO sites which found prostate-specific antigen levels in cervical samples to be less frequent in participants allocated to DMPA-IM than to the LNG implant and the Cu IUD [20]. However, there is a dearth of robust data regarding the comparative menstrual, psychological and behavioral effects of DMPA-IM and NET-EN.

Apart from direct exogenous effects of hormonal contraceptives (HCs), benefits and side-effects may be mediated by the effects of hormonal contraception on endogenous sex steroid hormones. Circulating levels of sex hormones modify cellular morphology in the brain [21] and influence higher brain functions such as cognition, memory and mood [22]. However, the neuropsychological and behavioral effects of sex hormones are complex and poorly understood. There is general agreement that estradiol levels are directly related to sexual desire [23], while there is some evidence that decreased estradiol levels are linked to depression [24]. Reduced estradiol levels by progestin-only contraceptives is emerging as a likely key factor influencing HIV susceptibility [12, 13, 25–28]. Normal estradiol levels in premenopausal individuals not on HC are generally associated with health benefits, while relatively low estradiol levels have the potential to exert multiple adverse effects, including on brain and cardiovascular function, lipid profiles, bone metabolism and bone mineral density, and on the female genital tract [12, 29–31]. The latter likely include effects on vaginal microbiome composition, genital tract integrity, immune function and susceptibility to and transmission of HIV and other infections [12, 13]. DMPA-IM use is associated with lower estrogen levels compared with no HC, as well as users of other HCs such as the LNG intrauterine system, and the etonogestrel and LNG implants [12, 13, 32]. Some reports suggest that DMPA-IM results in hypoestrogenic effects with estradiol levels similar to postmenopausal levels [13, 33]. NET-EN has also been reported to result in hypoestrogenic effects, albeit less so than for DMPA-IM users [13, 34], while other studies report NET-EN users having estradiol levels remaining in the normal premenopausal range [13, 35]. Inter-individual and inter-study values reported for estradiol differ greatly for the same contraceptive methods and are limited by low participant numbers [36]. Whether there are significant differences in estradiol levels between contraceptive methods, or whether these are confounded by differences in sampling times, differences in study participant numbers, demographic characteristics of the study populations and/or different methodologies for estradiol detection, is unclear from the literature. The extent to which DMPA-IM and NET-EN individually result in hypoestrogenism and their relative effects are unclear and are potentially crucial to understanding their individual and relative side-effects.

In settings where women prefer injectable contraceptives, more data is required to provide robust evidence to inform clinicians, policy-makers and participants about the individual and relative risks and benefits of NET-EN and DMPA-IM. Observational studies are fundamentally flawed in that unmeasurable personal characteristics may influence the choice of

contraceptive method and bias the results [5]. Here we have investigated the effects on estradiol levels and menstrual, psychological and behavioral effects of DMPA-IM and NET-EN within the context of a randomized open-label trial, the Women's Health, Injectable Contraception and HIV (WHICH) study.

## Methods

### Aim, design and setting

The primary aim of the WHICH study was to investigate the effects of DMPA-IM and NET-EN on estradiol levels and depression. Secondary aims included other hormonal effects, sexual behavioral, menstrual and immune effects within and between the two products. In this paper we report on the primary outcomes as well as some of the secondary outcomes, namely sexual behavioral and menstrual effects; and whether these measures are associated with estradiol levels. Towards this goal we conducted a parallel, open label, individually-randomized trial at the East London and Mdantsane public health clinics and hospitals (Frere and Cecilia Makiwane Hospitals), South Africa (331 participants), and the research site of MatCH Research Unit (MRU), University of the Witwatersrand, based in Durban, KwaZulu-Natal, South Africa (189 participants). A summarized protocol is available at https://pactr.samrc.ac. za/TrialDisplay.aspx?TrialID=6073.

### Participants

We recruited participants attending family planning clinics and those in the local communities who requested injectable contraception and intended to continue contraception for at least 18 months; were aged 18 to 40 years; legally competent to sign consent according to local regulations; prepared to use either DMPA-IM or NET-EN; prepared to accept follow-up procedures and able to fulfil these procedures, including routine HIV tests according to national guidelines; who after full counselling declined to use pre-exposure prophylaxis (PrEP) for HIV; understood the patient information form and signed written informed consent. Exclusion criteria were participants who had received DMPA-IM in the previous 6 months or NET-EN in the previous 4 months; were HIV positive; were planning to move out of the study area in the next 18 months; were participating in another clinical trial; were <6 weeks postpartum or post-abortion; had diabetes or high blood pressure; did not meet the WHO medical eligibility criteria (MEC) or local national guidelines for DMPA-IM or NET-EN use; or were using or intending to use medication which might have interfered with biological measurements such as steroids or drugs affecting renal function such as PrEP. Prospective participants were fully informed about PrEP, and if interested in using PrEP, were referred to a local provider. Participants were recruited and followed from 5 November 2018 to 30 November 2019. Participants who, after recruitment, changed their minds and decided to access PrEP services, were to remain in the study. To our knowledge, none did. Participants who met the entry criteria were fully counselled and informed in their preferred language and invited to participate. Participants were counselled on HIV risk reduction including condom use.

Exclusion of pregnancy and clinical assessment for sexually transmitted infections or contra-indications to the contraceptives were conducted, and any illness or pregnancy detected was managed in the routine service.

### Randomisation and masking

Allocation lists were prepared independently by SA Medical Research Council (SAMRC) using computer-generated random sequence in balanced blocks of variable size, stratified by

study site. Those enrolling participants could not predict the randomization sequence. Participants who agreed to participate were entered onto a trial register and then randomized by accessing the online randomization REDCap module [37]. In the event of difficulty accessing the online service, a separate series of randomized allocations was available in sequentially numbered, sealed opaque envelopes, or by telephone back-up service. Participants and research staff administering treatments were not masked to group allocation. Those conducting outcome interviews were not aware of the group allocation of participants, but this could have become apparent during some interviews.

## Procedures

Baseline demographic, menstrual, psychological, and behavioral data were recorded before randomization. Each participant was assigned a unique participant trial identification number (PTID) and data were collected using the PTID. Authors did not have access to information that could identify individual participants during or after data collection. Baseline blood (up to 40ml venous blood), dried blood spots and genital tract samples (cervical cytobrush and lateral vaginal wall swabs) for ancillary future immunological and hormonal studies and archiving were collected. Blood samples were separated and the serum stored at -80˚C. Participants were allocated to receive DMPA-IM 150mg intramuscular 12-weekly or NET-EN 200mg intramuscular 8-weekly. Strategies to manage side-effects without method change were explored with participants. In the event of discontinuation of either method, alternative choices were offered to participants according to national contraception guidelines. Participants were asked to attend the research sites at the time of their repeat injections (8- or 12-weekly) to 24 weeks, and at 25 weeks to collect 7-day post-injection biological samples. A 28-day daily symptom and behavior diary (S1 Table) was initiated at 24 weeks. At the final study visit, the participants were re-counselled about their future contraceptive choices. Further contraceptive care was provided within the routine provincial health services. Biological samples were collected and questionnaires administered at 25 weeks and participants were offered an HIV test by study staff, in line with national guidelines. Participants received approved compensation for their time and costs for in-person visits (R250 for study visits and R100 for contraception-only provision visits). Every attempt was made to contact participants who did not return for follow up including repeated calls to participant's and alternative phone numbers, and where possible home visits (provided previously consented to). Participants who acquired HIV were referred for HIV care to local healthcare facilities. Those who had depressive symptoms were counselled and referred.

## Outcomes

The primary laboratory outcome was serum 17β estradiol, and the primary clinical outcome was depression score (Beck Depression Inventory—BDI-II).

Estradiol was measured at Neuberg Global Laboratories (Durban, KwaZulu Natal, South Africa) by a chemiluminescent microparticle immunoassay (ARCHITECT Estradiol B7K720, analytical sensitivity ≤ 10 pg/mL) on stored baseline and 25-week (7 days after the 24-week injection) serum samples. HIV assays (finger prick, rapid HIV test) were performed on site. Additional hormonal and immunological studies are in progress and will be reported separately. The BDI-II method was chosen to evaluate depressive symptoms. It has previously been validated and used in the same cultural context and translated into the local languages IsiXhosa and IsiZulu. English, IsiZulu and IsiXhosa versions were used. Verbal administration was utilised. The BDI-II has 21 items, and each item is rated on a four-point scale ranging from 0–3. The maximum total score is 63. According to the BDI-II manual, scores of 0–13 indicate no or

minimal depression, scores of 14–19 indicate mild depression, scores of 20–28 indicate moderate depression, and scores of 29–63 indicate severe depression [38]. The Arizona Sexual Experiences Scale (ASEX) was used to evaluate sexual function. This is a five-item rating scale with total scores ranging from 5–30. It has been validated to be independent of the presence of a coital partner and can therefore be used even when study participants are not coitally active. Questions 4 and 5 of this scale are not ranked if a participant has not engaged in sexual intercourse within a week of the interview. A structured questionnaire was used to assess other secondary psychological and behavioral parameters: feeling sad for no reason, no menstruation, painless menstruation, no sexual intercourse, never use a condom during intercourse, and decreased sexual desire. Participants were asked to prospectively complete a 28-day daily diary at home of symptoms and behavior, commencing on the day of their 24-week visit (S1 Table). Parameters measured in the daily diary were: characteristics of menstruation, sexual intercourse with steady or casual partner, condom use, feeling sad for no reason, feeling the urge to have sexual intercourse, and feeling that partner loves her. The trial was not powered for HIV acquisition or pregnancy, but these were measured to provide an incidence estimate to inform a potential future larger trial.

## Statistical analysis

The primary laboratory outcome was serum estradiol. In a previous study [35], estradiol levels in postpartum participants randomized to NET-EN were 136 pmol/L (standard deviation (SD) 119). Using a two-sample Student's t-test and assuming a common SD of 119 pmol/L, a sample size of 181 participants per group was required to show a difference of 35 pmol/L in either direction with 95% certainty and 80% power. To account for a 15% loss to follow-up, we aimed to recruit 213 participants per group (http://pharmaschool.co/size4.asp).

The primary clinical outcome was depression score. In a previous study (35), the Montgomery-Asberg Depression Rating Scale (MADRS) scores in postpartum participants randomized to NET-EN were 8.3 (standard deviation (SD) 7.5).

Using a two-sample Student's t-test and assuming a common SD of 7.5, a sample size of 221 participants per group was required to show a difference of 2 with 95% certainty and 80% power. To account for 15% loss to follow-up, we aimed to recruit 260 participants per group (http://pharmaschool.co/size4.asp).

All measured clinical outcomes were reported, and secondary outcome comparisons were regarded as exploratory.

Statistical analysis was by intention to treat (ITT), and the results are reported according to the CONSORT guidelines. Data were analysed using Stata 16 (StataCorp, College Station, TX, USA). Descriptive statistics are presented as frequencies with percentages and means with standard deviations (SD). Where data was non-normally distributed, medians with interquartile ranges (IQR) are presented. Associations between categorical variables were assessed using Pearson's chi-squared test, or Fisher's exact test where applicable. Means were compared across arms using the Student's t-test and across timepoints using the paired Student's t-test. Medians were compared across arms using the Wilcoxon rank-sum test and across timepoints using the Wilcoxon matched-pairs signed-rank test. Median serum estradiol was compared across arms and across time points using a mixed-effects linear regression with random effects for site and participant and a diagonal covariance structure (selected on the assumption that levels within site and participant are not correlated). Through the incorporation of random effects, the mixed-effects linear regression model is able to account for the hierarchical structure of the data with participants being clustered by site and the repeated estradiol measurements clustered within participant. Risk/Rate ratios (RR) for differences between arms and

timepoints were estimated using generalised linear models with site and participant as a random effects and robust standard errors. Spearman correlation coefficients were calculated between serum estradiol and clinical and behavioral outcomes. Results were considered significant for p < 0.05. All analysis was conducted using Stata version 16 (StataCorp. 2019. Stata Statistical Software: Release 16. College Station, TX: StataCorp LLC).

### Ethics approval and consent

A feasibility study has shown that random allocation to different contraceptive methods is acceptable to most women [39]. Ethical approval was obtained from the Faculty of Health Sciences Human Research Ethics Committee (FHS HREC, M180528) of the University of Witwatersrand, and from the East London Hospital Institutional Ethics Committee. Permission to conduct the study was obtained from the Provincial Departments of Health of Eastern Cape and KwaZulu-Natal. The study was registered with the Pan African Clinical Trials Registry number PACTR 202009758229976. In 2020 the authors discovered that the initial online trial registration on the Pan African Clinical Trials Registry website had not been logged onto the system, due to missing information on individual participant data sharing. The original information plus individual participant data sharing statement were re-entered and approved on 1 September 2020: Trial number PACTR202009758229976. The authors confirm that all ongoing and related trials for this drug/intervention are registered. This study adhered to the ethical principles outlined in the Declaration of Helsinki (World Medical Association, 2011) and the Constitution of the Republic of South Africa (Bill of Rights). Written informed consent was obtained from all women to participate in the WHICH study. Research staff with Good Clinical Practice certification and specific training in the recruitment procedures conducted recruitment. Informed consent complied with requirements for research on human subjects. Both sites had active Community Advisory Boards who approved the study at the planning stage.

## Results

We screened 546 and randomized 521 participants between 5 November 2018 and 30 November 2019, 262 to DMPA-IM and 259 to NET-EN. The trial profile is shown in Fig 1. A total of 86.9% (n = 453) completed the 25-week study visit with a similar number completing in both method groups.

Baseline data are shown in Table 1 and were similar between groups. The mean age was 25 years in both groups and only 2% of participants were married. Most participants were unemployed, which is typical of the low-income populations we serve.

Overall, 92.8% (DMPA-IM, 206/222) and 92.9% (NET-EN, 209/225) of participants included in the hormonal analyses received all contraceptive injections over the study period. At 24 weeks, 94.3% (DMPA-IM, 211/222) and 98.3% (NET-EN, 222/225) of participants included in the hormonal analyses, received their randomised contraceptive.

Since the interaction effect between arm and time point were not significant (p = 0.866), we present the tests for the main effects in Table 2. Median estradiol levels at 25 weeks were 60% and 62% lower than at baseline for DMPA-IM and NET-EN, respectively (p<0.001). The median decreases were not statistically different between the groups (p = 0.467). The level at 25 weeks was 9% lower with NET-EN than DMPA-IM, which was not a statistically significant difference (DMPA-IM n = 222, median 76.5 (interquartile range 54.1 to 104.2) versus NET-EN n = 225, 69.8 (55.1 to 89.3 pmol/L), p = 0.450).

Among 6 dichotomous parameters (Table 3), estradiol levels were significantly greater among those reporting never using a condom at 25 weeks.

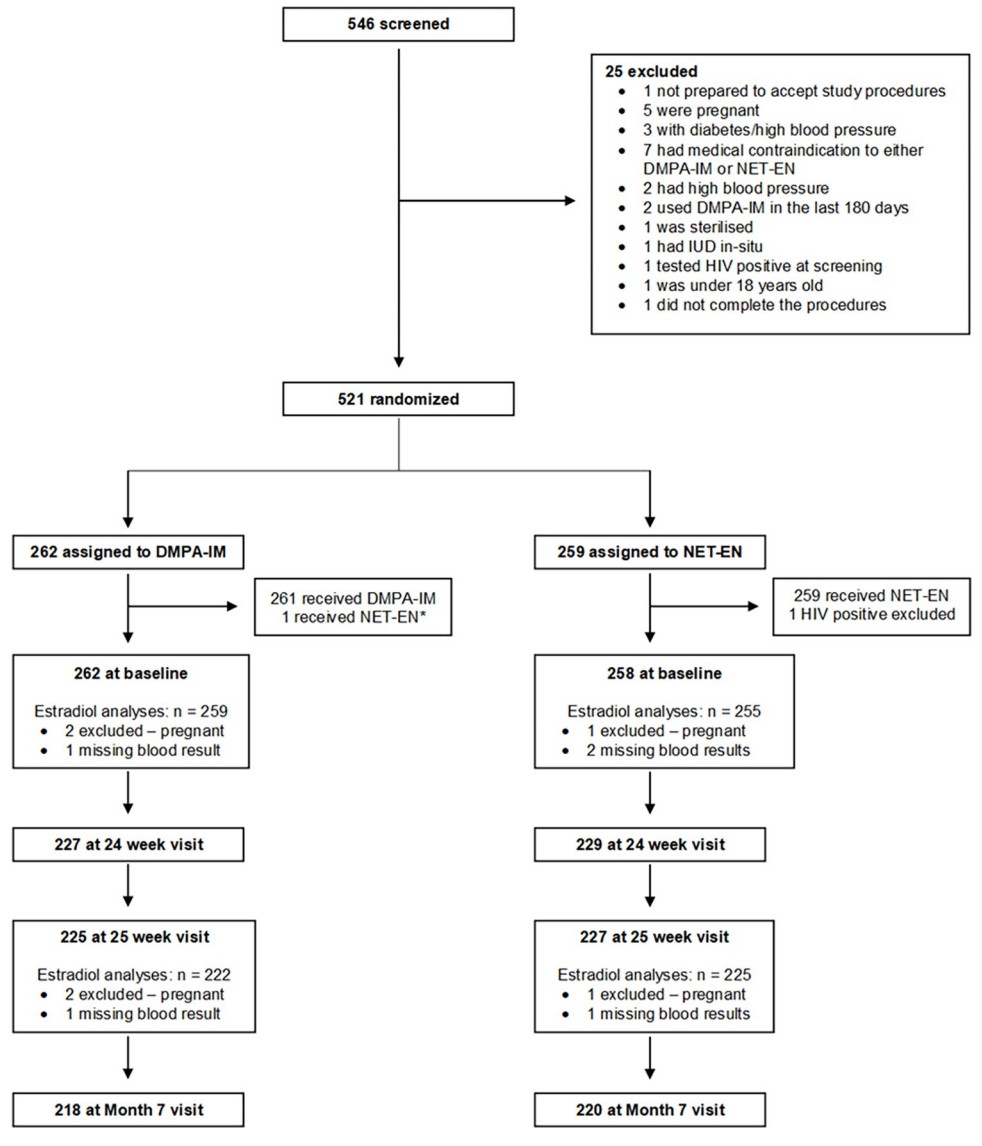

**Fig 1. Trial profile.**

There were no significant associations between estradiol levels and the daily diary outcomes 'feeling sad for no reason' and 'feeling partner does not love her' (Table 4).

Clinical results for baseline and 25 weeks are shown in Table 5. Four of 10 parameters were significantly changed from baseline to 25 weeks with both DMPA-IM and NET-EN: depression scores were lower, while feeling sad for no reason, amenorrhoea and never using a condom were increased at 25 weeks. In addition, the number with a BDI-II score indicating depression was significantly decreased at 25 weeks in only the DMPA-IM group and the report of decreased sexual desire was significantly increased in only the NET-EN group. No participants were found to have a BDI-II score indicating depression at 25 weeks.

To obtain more information about psychological behavior relevant to depression, sexual behavior and menstrual effects, in addition to the data in Table 5 for baseline and at 25 weeks, a 28-day daily diary was initiated at 24 weeks. This novel strategy was used to overcome potential limitations of data based on recall. The effects of DMPA-IM and NET-EN on depression,

**Table 1. Baseline characteristics of women by randomized method[a].**

| | DMPA-IM | | NET-EN | |
|---|---|---|---|---|
| | **n** | | **n** | |
| **Age (years)** | 262 | 24.9 (4.8) | 258 | 24.7 (4.6) |
| **Ethnicity** | 262 | | 258 | |
| Xhosa | | 176 (67.2) | | 167 (64.7) |
| Zulu | | 81 (30.9) | | 91 (35.3) |
| Mixed race | | 1 (0.4) | | 0 (0) |
| Other African ethnicity | | 4 (1.5) | | 0 (0) |
| **Marital status** | 262 | | 258 | |
| Single | | 256 (97.7) | | 252 (97.7) |
| Married | | 6 (2.3) | | 6 (2.3) |
| **Highest level of education** | 262 | | 258 | |
| Primary school, complete | | 3 (1.2) | | 4 (1.6) |
| High school, not complete | | 103 (39.3) | | 86 (33.3) |
| High school, complete | | 97 (37.0) | | 117 (45.4) |
| Post high school education | | 59 (22.5) | | 51 (19.8) |
| **Source of income** | 262 | | 258 | |
| Unemployed | | 220 (84.0) | | 224 (86.8) |
| Self-employed | | 5 (1.9) | | 3 (1.2) |
| Employed | | 37 (14.1) | | 31 (12.0) |
| **Previous use of method** | | | | |
| DMPA-IM | 262 | 193 (73.7) | 258 | 182 (70.5) |
| NET-EN | 262 | 84 (32.1) | 258 | 74 (28.7) |
| **Sexual dysfunction (ASEX)** | 262 | 15 (5.7) | 258 | 10 (3.9) |
| **Depression (BDI)** | 262 | 4 (1.5) | 258 | 1 (0.4) |
| **Feeling sad for no reason[b]** | 261 | 10 (3.8) | 255 | 8 (3.1) |
| **No menstruation[b]** | 260 | 34 (13.1) | 256 | 29 (11.3) |
| **Painless menstruation[b]** | 226 | 204 (90.3) | 227 | 202 (89.0) |
| **No sexual intercourse[b]** | 261 | 41 (15.7) | 256 | 53 (20.7) |
| **Never use a condom during intercourse[b]** | 220 | 24 (10.9) | 203 | 15 (7.4) |
| **Decreased sexual desire[b]** | 261 | 10 (3.8) | 256 | 10 (3.9) |

[a]Expressed as mean values (standard deviation) or n-value (percent)

[b]Any occurrence in the last 3 months.

sexual behavior and menstrual effects were assessed from the combined data in Tables 5 and 6. Among 28 clinical parameters which were compared (Tables 5 and 6), 10 were significantly different between DMPA-IM and NET-EN at 25 weeks. Participants allocated to DMPA-IM had more amenorrhoea (41.7 vs 37.2%, RR 1.12, 95% CI 1.02 to 1.23, p = 0.016) and fewer reported no sexual intercourse since the previous visit (19.3 vs 24.8%, RR 0.78, 95% CI 0.68 to 0.89, p <0.001) and on the daily diary (21.9 vs 24.9%, RR 0.89, 95% CI 0.88 to 0.90, p<0.001). Furthermore, on the daily diary, with DMPA-IM more sexual acts were reported per 28 days (median 4, IQR 1 to 10, vs 4, 1 to 8, RR 1.14, 1.10 to 1.18, p<0.001); more participants reported at least one act of unprotected coitus (56.7 vs 47.9%, RR 1.18, 95% CI 1.08 to 1.29, p <0.001); less condom use with steady partner was reported (median 14.3%, IQR 0 to 100 vs 47.2%, 0 to 100, p = 0.012); fewer days with urge for sexual intercourse (median 4, IQR 1 to 10 vs 4, 0 to 10, RR 0.97, 95% CI 0.96 to 0.99, p = 0.004); fewer days feeling partner does not love her (median 0, IQR 0 to 1 vs 0, 0 to 3, RR 0.58, 95% CI 0.34 to 0.98, p = 0.044); and fewer days

**Table 2. Estradiol (pmol/L) levels at baseline and 25 weeks[a].**

| Estradiol | DMPA-IM | | | NET-EN | | | DMPA-IM vs NET EN |
|---|---|---|---|---|---|---|---|
| | n | Median | IQR | n | Median | IQR | p-value[b] |
| Baseline | 259 | 189.4 | 113.9–401 | 255 | 183.2 | 107.1–382.7 | 0.787 |
| 25 weeks | 222 | 76.5 | 54.1–104.2 | 225 | 69.8 | 55.1–89.3 | 0.450 |
| Change from baseline | 221 | -114.1 | -316.9 –-38.8 | 225 | -99.3 | -280.2 –-31.1 | 0.467 |
| Change from baseline p-value[c] | < 0.001 | | | < 0.001 | | | |

[a]Differences expressed as p-values, ITT analysis.

[b]Mixed-effects linear regression accounting for clustering by site.

[c]Mixed-effects linear regression accounting for clustering by site and participant.

feeling sad for no reason (median 1, IQR 0 to 5 vs 1, 0 to 6, RR 0.80, 95% CI 0.68 to 0.95, p = 0.011). The study was not powered to compare pregnancy rate, which occurred in 2 vs 1 participants, nor HIV acquisition which occurred in 4 vs 6 participants allocated to DMPA-IM vs NET-EN, respectively.

Among all participants from both groups, there was a significant weak negative correlation between the primary hormonal outcome (estradiol) and the primary clinical outcome (depression–BDI-II) both at baseline (p = 0.026) and at 25 weeks (p = 0.004) (S2 Table).

There was a weak positive correlation between estradiol and sexual dysfunction (ASEX) which was significant at 25 weeks (p = 0.018) but not at baseline (p = 0.054). There was no significant correlation of estradiol levels with daily diary recording of number of sexual acts, number of condomless sexual acts or number of days with urge for sexual intercourse.

**Table 3. Estradiol (pmol/L) association with physiological, psychological and behavior[a].**

| | Baseline | | | | 25 weeks | | | |
|---|---|---|---|---|---|---|---|---|
| | n | Median | IQR | p-value | n | Median | IQR | p-value[c] |
| **Feeling sad for no reason[b]** | | | | 0.064 | | | | 0.809 |
| No | 492 | 185.1 | 109.2–382 | | 378 | 73.8 | 54.1–96.0 | |
| Yes | 18 | 358.0 | 193.2–512.7 | | 67 | 69.8 | 55.8–91.3 | |
| **No menstruation[b]** | | | | 0.248 | | | | 0.888 |
| No | 447 | 201.7 | 110.2–400.0 | | 272 | 73.3 | 55.1–94.3 | |
| Yes | 63 | 163.8 | 113.4–306.5 | | 174 | 74.0 | 54.1–99.7 | |
| **Painless menstruation[b]** | | | | 0.377 | | | | 0.109 |
| No | 46 | 270.0 | 144.4–429.9 | | 30 | 75.7 | 52.3–117.0 | |
| Yes | 401 | 196.5 | 107.6–384.4 | | 242 | 72.3 | 55.1–93.9 | |
| **No sexual intercourse[b]** | | | | 0.110 | | | | 0.191 |
| No | 417 | 190.8 | 110.5–398.4 | | 347 | 73.1 | 54.1–95.1 | |
| Yes | 94 | 178.7 | 116.5–321.8 | | 99 | 76.1 | 57.3–103.1 | |
| **Never use a condom during intercourse[b]** | | | | 0.515 | | | | 0.003 |
| No | 379 | 195.5 | 110.2–400.0 | | 276 | 71.3 | 52.9–92.2 | |
| Yes | 38 | 164.6 | 114.6–382.1 | | 71 | 79.4 | 61.0–106.5 | |
| **Decreased sexual desire[b]** | | | | 0.158 | | | | 0.737 |
| No | 492 | 189.3 | 113.4–397.8 | | 424 | 73.6 | 54.9–97.8 | |
| Yes | 19 | 159.2 | 89.9–269.3 | | 19 | 76.7 | 53.6–85.9 | |

[a]Based on the questionnaire responses.

[b]Any occurrence in the last 3 months.

[c]Mixed-effects linear regression accounting for clustering by site

**Table 4. Association between 25-week estradiol (pmol/L) levels and 28-day Daily Diary results.**

| | No days | | | At least one day | | | p-value[a] |
|---|---|---|---|---|---|---|---|
| | n | Median | IQR | n | Median | IQR | |
| **Feeling sad for no reason** | | | | | | | |
| Estradiol | 164 | 74.8 | 58.5–94.9 | 258 | 73.0 | 53.5–97.0 | 0.508 |
| **Feeling partner does not love her** | | | | | | | |
| Estradiol | 287 | 73.3 | 55.1–96.5 | 135 | 74.4 | 55.0–93.9 | 0.812 |

[a]Mixed-effects linear regression accounting for clustering by site.

**Table 5. Clinical outcomes at baseline and 25 weeks[a].**

| | DMPA-IM | | | | | | NET-EN | | | | | | DMPA-IM vs NET-EN | |
|---|---|---|---|---|---|---|---|---|---|---|---|---|---|---|
| | Baseline | | 25 w | | 25 vs baseline | | Baseline | | 25 w | | 25 vs baseline | | 25 weeks | |
| | n | Results[a] | n | Results[a] | RR (95% CI) | p-value | n | Results[a] | n | Results[a] | RR (95% CI) | p-value | RR (95% CI) | p-value |
| **Depression (BDI)** | 262 | 4 (1.5) | 224 | 0 (0) | - | 0.046 | 258 | 1 (0.4) | 226 | 0 (0) | - | 0.317 | - | - |
| **BDI score** | 262 | 1 (0–3) | 224 | 1 (0–2) | - | < 0.001 | 258 | 2 (0–3) | 227 | 0 (0–2) | - | < 0.001 | - | 0.950 |
| **Sexual dysfunction (ASEX)** | 262 | 15 (5.7) | 224 | 9 (4.0) | 0.70 (0.18–2.71) | 0.611 | 258 | 10 (3.9) | 226 | 7 (3.1) | 0.81 (0.21–3.21) | 0.769 | 1.30 (0.36–4.64) | 0.689 |
| **ASEX score** | 262 | 8.9 (3.9) | 224 | 8.9 (3.9) | - | 0.749 | 258 | 8.9 (3.6) | 226 | 8.9 (3.7) | - | 0.974 | - | 0.911 |
| **Feeling sad for no reason[b]** | 261 | 10 (3.8) | 222 | 34 (15.3) | 5.02 (1.32–19.03) | 0.018 | 255 | 8 (3.1) | 226 | 35 (15.5) | 2.29 (1.19–4.40) | 0.013 | 0.99 (0.78–1.26) | 0.936 |
| **No menstruation[b]** | 260 | 34 (13.0) | 223 | 93 (41.7) | 3.19 (2.25–4.53) | < 0.001 | 256 | 29 (11.2) | 226 | 84 (37.2) | 3.28 (3.05–3.54) | < 0.001 | 1.12 (1.02–1.23) | 0.016 |
| **Painless menstruation[b]** | 226 | 204 (77.9) | 130 | 114 (87.7) | 0.97 (0.82–1.15) | 0.731 | 227 | 202 (78.3) | 142 | 128 (90.1) | 1.01 (0.76–1.36) | 0.932 | 0.97 (0.94–1.01) | 0.138 |
| **No sexual intercourse[b]** | 261 | 41 (15.7) | 223 | 43 (19.3) | 1.23 (0.95–1.58) | 0.119 | 256 | 53 (20.5) | 226 | 56 (24.8) | 1.20 (0.87–1.65) | 0.270 | 0.78 (0.68–0.89) | < 0.001 |
| **Never use a condom during intercourse[b]** | 220 | 24 (9.2) | 180 | 44 (24.4) | 2.24 (1.78–2.82) | < 0.001 | 203 | 15 (5.8) | 170 | 30 (17.7) | 2.39 (1.45–3.93) | 0.001 | 1.39 (0.78–2.47) | 0.270 |
| **Decreased sexual desire[b]** | 261 | 10 (3.8) | 222 | 9 (4.0) | 2.05 (0.48–8.79) | 0.234 | 256 | 10 (3.9) | 224 | 10 (4.4) | 3.10 (1.97–4.88) | < 0.001 | 0.91 (0.79–1.05) | 0.191 |
| **Pregnant** | | | 223 | 2 (0.9) | | | | | 226 | 1 (0.4) | | | 2.03 (0.04–104.0) | 0.725 |
| **HIV positive** | | | 224 | 4 (1.8) | | | | | 226 | 6 (2.7) | | | 0.67 (0.05–9.45) | 0.769 |

[a]Results expressed as n-value (percent), mean (standard deviation) or median (interquartile range). Differences expressed as risk/rate ratios (RR) with 95% confidence intervals (CI) and p-values, ITT analysis.

[b]Any occurrence in the last 3 months.

**Table 6. 28-Day daily diary data[a].**

| | DMPA-IM | | NET-EN | | DMPA-IM vs NET-EN | |
|---|---|---|---|---|---|---|
| | n | Results[a] | n | Results[a] | RR (95% CI) | p-value |
| **No menstruation** | 210 | 149 (71.0%) | 217 | 154 (71.0%) | 1.00 (0.93–1.07) | 0.995 |
| **Menstruation duration in days** | 61 | 2 (1–5) | 63 | 4 (1–6) | 0.92 (0.74–1.16) | 0.487 |
| **At least one day of severe menstrual pain** | 61 | 6 (9.8%) | 63 | 6 (9.5%) | 1.05 (0.56–1.98) | 0.877 |
| **No sexual intercourse** | 210 | 46 (21.9%) | 217 | 54 (24.9%) | 0.89 (0.88–0.90) | < 0.001 |
| **Sexual acts per 28 days** | 210 | 4 (1–10) | 217 | 4 (1–8) | 1.14 (1.10–1.18) | < 0.001 |
| **At least one act of unprotected intercourse** | 210 | 119 (56.7%) | 217 | 104 (47.9%) | 1.18 (1.08–1.29) | < 0.001 |
| **Number of days with unprotected intercourse** | 210 | 2 (0–6) | 217 | 0 (0–4) | 1.29 (1.08–1.53) | 0.004 |
| **Condom use with casual partners (as percentage of sexual intercourse with a casual partner days)** | 20 | 100 (40–100) | 28 | 74.1 (0–100) | - | 0.123 |
| **Condom use with steady partners (as percentage of sexual intercourse with steady partner days)** | 160 | 14.3 (0–100) | 156 | 47.2 (0–100) | - | 0.012 |
| **Number of women with at least one occurrence of intra-menstrual coitus** | 210 | 14 (6.7%) | 217 | 17 (7.8%) | 0.85 (0.53–1.37) | 0.507 |
| **Number of days with intra-menstrual coitus** | 14 | 1.5 (1–2) | 17 | 1 (1–2) | 1.05 (0.93–1.19) | 0.410 |
| **Number of women with at least one occurrence of condomless intra-menstrual coitus** | 210 | 10 (4.8) | 217 | 6 (2.8) | 1.72 (0.74–4.02) | 0.209 |
| **Number of days with condomless intra-menstrual coitus** | 10 | 1 (1–2) | 6 | 1 (1–3) | 0.78 (0.46–1.35) | 0.381 |
| **Number of days with urge for sexual intercourse** | 210 | 4 (1–10) | 217 | 4 (0–10) | 0.97 (0.96–0.99) | 0.004 |
| **Number of women with at least one day feeling partner does not love her** | 210 | 59 (28.1%) | 217 | 78 (35.9%) | 0.78 (0.49–1.23) | 0.279 |
| **Number of days feeling partner does not love her** | 210 | 0 (0–1) | 217 | 0 (0–3) | 0.58 (0.34–0.98) | 0.044 |
| **Number of women with at least one day feeling sad for no reason** | 210 | 127 (60.5%) | 217 | 135 (62.2%) | 0.97 (0.85–1.12) | 0.692 |
| **Number of days feeling sad for no reason** | 210 | 1 (0–5) | 217 | 1 (0–6) | 0.80 (0.68–0.95) | 0.011 |

[a]Results expressed as n-value (percent), mean (standard deviation) or median (interquartile range). Differences expressed as risk/rate ratios (RR) with 95% confidence intervals (CI) and p-values, ITT analysis.

## Discussion

We found a substantial and similar reduction in estradiol levels to postmenopausal levels with DMPA-IM and NET-EN. Compared with DMPA-IM, NET-EN users reported significantly less amenorrhoea, less sexual activity, more condom use, and more feeling sad and unloved. These behavioral outcomes suggest less sexual exposure with NET-EN. The strength of this evidence is high due to the randomized study design and the consistency of results across the outcomes measured.

This study is the first randomized trial to compare the effects of DMPA-IM and NET-EN on menstrual, psychological and behavioral measures and levels of estradiol. No prior

randomized information is available, to our knowledge, regarding relative effects of DMPA-IM and NET-EN use on depression. While one [40] but not another study [19] reports that DMPA-IM users report higher coital frequency and lower condom usage compared to other contraceptive methods, no information appears to be available for effects of NET-EN on sexual behavior. Although previous observational studies have reported that both DMPA-IM and NET-EN result in reduced levels of estradiol [12, 13, 33–36], there is no robust data on their relative effects. There is also a lack of clarity on the relationship between sex hormones and psychological wellbeing. In a comprehensive systematic review, estradiol levels were found to be lower in participants with premenstrual dysphoric disorder and postpartum depression, but not perimenopausal depression nor depression unrelated to reproductive transition phases [41]. A 2020 systematic review concluded that "associations between endogenous sex hormones and depressive symptoms were inconclusive" and called for further research [42]. Our randomized study addresses these gaps in the literature regarding the method-specific and comparative effects DMPA-IM and NET-EN use.

The weak negative correlation between estradiol levels and BDI-II score in our study supports the role of low estradiol levels in depression in young individuals. This is consistent with our findings of an increase in feeling sad for both contraceptives (Table 5) and estradiol suppression at 25 weeks. However, at 25 weeks the depression scores as assessed by BDI-II in both groups were lower than at baseline. This may be a reflection of a broader assessment of wellbeing with the BDI, reflecting a general sense of wellbeing associated with the supportive environment and excellent care typical of a research setting. It has also been hypothesized that individuals may respond differently to estradiol changes in either direction with respect to depressive symptoms [43]. The depression scores at 25 weeks were too low for meaningful statistical comparison between groups.

We investigated for the first time the relationship between estradiol levels and sexual behavior in premenopausal women randomly allocated to DMPA-IM and NET-EN. The weak positive correlation suggesting an association between low estradiol levels and a low ASEX score (indicative of normal sexual activity and low sexual disfunction) (only at 25 weeks) in healthy young women in our study was unexpected. Contrary to our results, estradiol is considered to play a positive role in sexual desire and arousal in premenopausal women [44]. Consistent with this, sexual function has been found to deteriorate with decreasing ovarian function, and to be improved by hormone replacement therapy with 'natural' estrogen [45]. It is possible that our data are not due to a causal relationship but that low estrogen levels may be associated with changes in the levels of other hormones which may be affecting sexual behavior.

Among several behavioral measures, those that were significantly different between DMPA-IM and NET-EN at 25 weeks consistently indicated less coital activity and less condomless coital activity with NET-EN than with DMPA-IM. This might be a direct differential effect of the two progestins, or secondary to the differential effect on amenorrhoea (more menstruation-related avoidance of coitus in the NET-EN group).

The relationship between menstruation, coital exposure and HIV risk is complex. On the one hand, reduced menstruation may increase coital exposure overall and thus HIV risk. An in-depth interview study among Malawian women using progestogen contraception reported that some women ascribed their partner's infidelity to their partner's disinterest in sex with them during menstrual or breakthrough bleeding [46]. On the other hand, increased menstruation may be associated with more coitus during menstruation and thus with greater HIV risk [47, 48].

Estradiol levels were profoundly suppressed (at least by 60%) at 25 weeks with both injectables, reaching postmenopausal levels. The effects were not significantly different between groups. Our findings of estradiol serum concentrations for DMPA-IM of 76.5 pmol/L (IQR

54.1 to 104.2) are similar to values reported in the literature, which range from 37–367 pmol/L [32, 36, 49–54]. However, our findings of estradiol serum concentrations for NET-EN of 69.8 pmol/L (55.1 to 89.3), are lower than some but not all of those usually reported, which range from 135–2820 pmol/L [35, 55–57]. Possible discrepancies with the literature may be due to time of sampling, which is often not defined, or measured at lower progestin levels, just before the next injection, while our sampling was done at about one week post injection, which should correspond to near peak serum contraceptive levels.

Our data from a randomized trial analysing estrogen levels for at least 222 participants at both baseline and 1 week after the last injection at 6 months are the most definitive and robust results to date on the individual and relative effects of DMPA-IM and NET-EN on estrogen levels. Although a wide range of estrogen levels are reported for premenopausal women ranging from 149–1930 pmol/L [58–61], and for postmenopausal participants from 22–161.5 pmol/L [58, 62, 63], the estrogen values from our study at peak MPA and NET serum levels are more similar to estrogen levels in postmenopausal women. Furthermore, our findings that both DMPA-IM and NET-EN repress estrogen by at least 60% and that this degree of repression is not significantly different between arms is highly relevant to their potential individual and relative side-effects. Whether estrogen levels of DMPA-IM and NET-EN fluctuate depending on time after injection, and/or are affected by number of injections, remains to be determined.

The secondary outcome results should be interpreted with caution because of multiple comparisons. A cautious interpretation is that the consistently lower coital and unprotected coital exposure with NET-EN than with DMPA-IM in several measures is consistent with lower HIV exposure with NET-EN than with DMPA-IM.

As was found in the ECHO trial [19], we found a high rate of HIV seroconversion in a cohort of young women who received consistent, comprehensive counselling on HIV prevention (10 seroconversions among 452 participants over 25 weeks).

## Limitations

Although an inclusion criterion for enrolment in the WHICH study was no injectable contraception in the last 4 months (NET-EN) or 6 months (DMPA-IM), this was ascertained via self-report and not verified biologically. Discrepancies between self-reported and biologically confirmed prior contraceptive exposure have been reported in other studies [64]. In addition, use of oral contraception was permitted up to the day preceding enrolment. A total of 35 women reported using oral contraceptives, most recently 24 days before enrolment, with no difference between arms. It is therefore likely that some participants in both groups had some residual estradiol suppression at baseline. In view of the robust randomization procedures, it is expected that such effects would be balanced between groups. For this reason, the estradiol suppression measured at 25 weeks is likely to be an underestimation of the true degree of suppression. We have included the baseline hormone levels to confirm comparability of the groups, and our primary comparison between groups is based on both absolute levels at 25 weeks and changes from baseline.

Our findings may not be generalizable to individuals with different personal characteristics, for example, older women.

## Conclusions

Differences in HIV risk between NET-EN and DMPA-IM might be mediated by multiple immunological, hormonal, behavioral and other mechanisms. Our findings suggest that if NET-EN has a lower risk of HIV acquisition relative to DMPA-IM as reported in observational

studies, then this is unlikely to be related to major differences in their hypoestrogenic effects. Our behavioral data are consistent with less coitus and condomless coitus with NET-EN than with DMPA-IM and thus possibly less HIV exposure. The significant associations between estradiol levels and BDI and ASEX scores suggest that estradiol levels may be an important biological factor, but given the similar hypoestrogenic effects, is unlikely to account for differences between groups. Alternative explanations, not investigated in this report, might include different androgenic effects of DMPA-IM and NET-EN. The results showing postmenopausal levels of estrogen for both contraceptives support the inclusion of an estrogen replacement component for progestin contraceptives.

We have shown that random allocation to these similar and popular contraceptive products is well accepted by most participants approached. A major, pragmatic randomized trial powered to compare HIV acquisition is eminently feasible.

## Supporting information

**S1 Checklist. CONSORT 2010 checklist of information to include when reporting a randomised trial*.**
(DOC)

**S1 Table. Daily diary questionnaire.**
(DOCX)

**S2 Table. Spearman correlations between estradiol (pmol/L) levels and BDI, ASEX and daily diary scores.**
(DOCX)

**S1 File.**
(PDF)

## Acknowledgments

We thank the participants who participated in the study and the research staff who conducted the study procedures.

## Author Contributions

**Conceptualization:** Mandisa Singata-Madliki, Jenni Smit, Janet P. Hapgood, G. Justus Hofmeyr.

**Data curation:** Mandisa Singata-Madliki, Jenni Smit, Mags Beksinska, Yusentha Balakrishna, Chanel Avenant, Ivana Beesham, Ishen Seocharan, Joanne Batting, Janet P. Hapgood, G. Justus Hofmeyr.

**Formal analysis:** Yusentha Balakrishna.

**Funding acquisition:** Mandisa Singata-Madliki, Janet P. Hapgood, G. Justus Hofmeyr.

**Investigation:** Mandisa Singata-Madliki, Jenni Smit, Mags Beksinska, Yusentha Balakrishna, Chanel Avenant, Ivana Beesham, Ishen Seocharan, Joanne Batting, Janet P. Hapgood, G. Justus Hofmeyr.

**Methodology:** Mandisa Singata-Madliki, Yusentha Balakrishna, Chanel Avenant, Ishen Seocharan, Janet P. Hapgood, G. Justus Hofmeyr.

**Project administration:** Mandisa Singata-Madliki, Jenni Smit, Mags Beksinska, Chanel Avenant, Ivana Beesham, Joanne Batting.

**Supervision:** Janet P. Hapgood, G. Justus Hofmeyr.

**Visualization:** Yusentha Balakrishna.

**Writing – original draft:** Mandisa Singata-Madliki, Janet P. Hapgood, G. Justus Hofmeyr.

**Writing – review & editing:** Mandisa Singata-Madliki, Jenni Smit, Mags Beksinska, Yusentha Balakrishna, Chanel Avenant, Ivana Beesham, Ishen Seocharan, Joanne Batting, Janet P. Hapgood, G. Justus Hofmeyr.

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
