## [Decision Letter · Decision Letter 0]

5 Jul 2023

PONE-D-23-10439Effects of injectable contraception with depot medroxyprogesterone acetate or norethisterone enanthate on estradiol levels and menstrual, psychological and behavioral measures relevant to HIV risk: the WHICH randomized trialPLOS ONE

Dear Dr. Beesham,

Thank you for submitting your manuscript to PLOS ONE. After careful consideration, we feel that it has merit but does not fully meet PLOS ONE’s publication criteria as it currently stands. Therefore, we invite you to submit a revised version of the manuscript that addresses the points raised during the review process.

ACADEMIC EDITOR: Pay attention to these questions and all the reviewer comments below

How was randomization concealed to ensure unpredictability of the subsequent arm assignment by those randomizing participants?What is the usefulness of p-value in table 1 for a randomized trial?Why >80% were unemployed? Was this systematic or came out of a random process?==============================

We look forward to receiving your revised manuscript.

Kind regards,

Andrew Max Abaasa, Ph.D.

Academic Editor

PLOS ONE

3. We note that the original protocol that you have uploaded as a Supporting Information file contains an institutional logo. As this logo is likely copyrighted, we ask that you please remove it from this file and upload an updated version upon resubmission.

Reviewers' comments:

Reviewer's Responses to Questions

**Comments to the Author**

1. Is the manuscript technically sound, and do the data support the conclusions?

Reviewer #1: Yes

Reviewer #2: Yes

2. Has the statistical analysis been performed appropriately and rigorously? 

Reviewer #1: Yes

Reviewer #2: Yes

3. Have the authors made all data underlying the findings in their manuscript fully available?

Reviewer #1: Yes

Reviewer #2: Yes

4. Is the manuscript presented in an intelligible fashion and written in standard English?

Reviewer #1: Yes

Reviewer #2: Yes

5. Review Comments to the Author

Reviewer #1: A two-arm randomized clinical trial was conducted to compare hormonal, behavioral, and menstrual effects between the two injectable contraceptives. At 25 weeks, estradiol levels were statistically significantly lower than at baseline for both contraceptives. No statistical difference was observed between the two methods. Behavioral outcomes suggest less sexual exposure with NET-EN than DMPA-IM.

Minor revisions:

1- Lines 277-288: State the statistical testing method which achieves 80% power. Perhaps the power estimates are based on Student’s t-tests.

2- Line 302: Indicate if the interaction effect of arm and time point was included. The results presented in Table 2 should be based on a test of the interaction effect. If the interaction effect is significant, provide an interpretation of the results, but do not test main effects because the tests for main effects are uninteresting in light of significant interactions. If the interaction effects are non-significant, drop the interaction effects from the model and test the main effects. Determining which results to present when testing interactions is often a multi-step process.

3- In the statistical analysis section, state and describe the use of mixed-effects linear regression. Also state the underlying covariance structure used and the criteria for selecting it.

4- Tables 5-6: Provide a more descriptive column label than “Stat.”

5- Cite the statistical software used for the analysis.

Reviewer #2: This paper is based on a open-label trial that provides valuable contributions to this area of research. This paper would benefit from clarifying the exposures, outcomes and aims of the paper. My understanding was the aim of the paper was to look at the impacts of DMPA-IM and NET-EN on hormonal (estradiol levels measured via immunoassay), psychological (depression score, via verbal administration), behavioural (Arizona Sexual Experiences Scale, questionnaire daily), and menstrual (structured questionnaire & daily diary) factors. Whilst this is what seems to be the aim, the results present much more broad results than this which makes it slightly confusing and feels like there is too much information trying to be displayed in one paper. For example I am not sure what aim table 3 or table 4 is supposed to be addressing.

The diaries are a really interesting aspect of this study but I feel like they get brushed over without much discussion - as said earlier it feels like too much data is trying to be reported in this paper and would be better if it focussed more specifically on certain data as opposed to trying to cover it all. For table 6, the data is quite confusing and it isn't intuitive what to take away from the table.

The paper mentions the 'significance level is <0.05', it would be good to explain the justification for this. Within epidemiology we are trying to shift away from this as an arbitrary cut off. Also moving away from data being deemed 'statistically significant' but focusing more on the strength of evidence.

On page 16 in the final paragraph it states 'The level (of estradiol) at 25 weeks was 9% lower with NET-EN than DMPA-IM', but then a few sentences later it states 'Median estradiol decreased by 14.8 pmol/L more in the DMPA arm than the NET-EN arm, a difference which was not statistically significant (p = 0.467)'. These two statements seem contradictory?

Table 2 - please check the numbers, the median change from baseline don't seem to add up.

I would suggest rewording the discussion so the key findings of this paper come first.

6. PLOS authors have the option to publish the peer review history of their article (what does this mean?). If published, this will include your full peer review and any attached files.

Reviewer #1: No

Reviewer #2: No

---

## [Author Response · Author response to Decision Letter 0]

26 Jul 2023

Responses to editor's and reviewers' comments uploaded as an attachment.

---

## [Decision Letter · Decision Letter 1]

29 Nov 2023

Effects of injectable contraception with depot medroxyprogesterone acetate or norethisterone enanthate on estradiol levels and menstrual, psychological and behavioral measures relevant to HIV risk: the WHICH randomized trial

PONE-D-23-10439R1

Dear Dr. Beesham,

We’re pleased to inform you that your manuscript has been judged scientifically suitable for publication and will be formally accepted for publication once it meets all outstanding technical requirements.

Kind regards,

Hanna Landenmark

Staff Editor

PLOS ONE

Additional Editor Comments (optional):

Reviewers' comments:

Reviewer's Responses to Questions

**Comments to the Author**

1. If the authors have adequately addressed your comments raised in a previous round of review and you feel that this manuscript is now acceptable for publication, you may indicate that here to bypass the “Comments to the Author” section, enter your conflict of interest statement in the “Confidential to Editor” section, and submit your "Accept" recommendation.

Reviewer #1: All comments have been addressed

Reviewer #3: All comments have been addressed

2. Is the manuscript technically sound, and do the data support the conclusions?

Reviewer #1: (No Response)

Reviewer #3: Yes

3. Has the statistical analysis been performed appropriately and rigorously? 

Reviewer #1: (No Response)

Reviewer #3: Yes

4. Have the authors made all data underlying the findings in their manuscript fully available?

Reviewer #1: (No Response)

Reviewer #3: Yes

5. Is the manuscript presented in an intelligible fashion and written in standard English?

Reviewer #1: (No Response)

Reviewer #3: Yes

6. Review Comments to the Author

Reviewer #1: (No Response)

Reviewer #3: Nicely done. My only suggestion to the authors for future such research is to initiate the diary notations immediately upon assignment of the products or shortly thereafter, rather than only after the 1st year. In observational studies and real world contexts, many women discontinue their contraceptives during this interval - although most of these women did not. It was a missed opportunity to obtain this information prospectively to complete a picture of users and especially in a comparative setting such as this trial with the 2 different formulas.

7. PLOS authors have the option to publish the peer review history of their article (what does this mean?). If published, this will include your full peer review and any attached files.

Reviewer #1: No

Reviewer #3: No

---

## [Editor Report · Acceptance letter]

17 Mar 2024

PONE-D-23-10439R1 

PLOS ONE

Dear Dr. Beesham, 

I'm pleased to inform you that your manuscript has been deemed suitable for publication in PLOS ONE. Congratulations! Your manuscript is now being handed over to our production team.

Kind regards, 

on behalf of

Dr. Hanna Landenmark 

Staff Editor

PLOS ONE